# Inversion of Wheat Leaf Area Index by Multivariate Red-Edge Spectral Vegetation Index

Xiaoxuan Wang [1] , Guosheng Cai [1,*], Xiaoping Lu [1], Zenan Yang [1], Xiangjun Zhang [2] and Qinggang Zhang [1]

1    Key Laboratory of Spatio-Temporal Information and Ecological Restoration of Mines of Natural Resources of the People's Republic of China, Henan Polytechnic University, Jiaozuo 454003, China
2    Institute of Remote Sensing and Surveying and Mapping, Zhengzhou 450000, China
*    Correspondence: 212004020039@home.hpu.edu.cn; Tel.: +86-391-3987708

**Abstract:** Leaf area index (LAI) is an important parameter that determines the growth status of winter wheat and impacts the ecological and physical processes of plants in ecosystems. The problem of spectral saturation of winter wheat LAI at the booting stage was easily caused by the inversion of the univariate red-edge spectral vegetation index constructed by the red-edge band. In this paper, a new method that the univariate red-edge spectral vegetation index constructed in the red-edge band is used to invert the spectral saturation of the winter wheat LAI. The multivariable red-edge spectral vegetation index is used to invert the winter wheat LAI. This method can effectively delay the phenomenon of spectral saturation and improve the inversion precision. In this study, the Sentinel-2 data were used to invert the winter wheat LAI. An univariate and multivariate red-edge spectral vegetation index regression model was constructed based on the Red-edge Normalized Difference Spectral Indices 1 (NDSI1), Red-edge Normalized Difference Spectral Indices 2 (NDSI2), Red-edge Normalized Difference Spectral Indices 3 (NDSI3), Modified Chlorophyll Absorption Ratio Index (MCARI), MERIS Terrestrial Chlorophyll Index (MTCI), Transformed Chlorophyll Absorption in Reflectance Index (TCARI), and Transformed Chlorophyll Absorption in Reflectance Index/the optimized soil adjusted vegetation index (TCARI/OSAVI). Based on the correlation coefficient, the coefficient of determination ($R^2$), the root mean square error (RMSE) and noise equivalent value (NE), the best model was selected and verified to generate an inverted map. The results showed that the multivariable red-edge spectral vegetation index of NDSI1 + NDSI2 + NDSI3 + TCARI/OSAVI + MCARI + MTCI + TCARI was the best model for inverting the winter wheat LAI. The $R^2$, the RMSE and the NE values were all satisfied the requirements of the inversion precision ($R^2$ = 0.8372/0.8818, RMSE = 0.2518/0.1985, NE = 5/5). In summary, this method can be used to judge the growth of winter wheat and provide an accurate basis for monitoring crop growth.

**Keywords:** red-edge band; multivariate red-edge spectral vegetation index; leaf area index; Sentinel-2; winter wheat

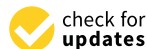



## 1. Introduction

Leaf area index (LAI) is half of the sum of all leaf areas on the surface area; however, it is one of the important agroecological parameters affecting the growth of the vegetation [1–3]. Studies have shown that the LAI is a key variable for monitoring and predicting the growth of the crop [4]. Therefore, accurate and rapid access to the LAI can provide reliable supporting information for governmental management of agricultural resources.

The method originally used to obtain the growth of crops is field investigation, which is time-consuming and not conducive to the implementation of region areas [5]. With the development of remote sensing, researchers have begun to estimate the growth of crops by the crop growth model and vegetation index regression model based on the remote sensing data. The crop growth models are mostly based on the growth mechanism of the crop and the structural parameters of the crop need to be input, which are complicated.

The vegetation index regression model mainly inverts LAI by the regression equation with measured LAI based on vegetation index, which are acquired by remote sensing data. The model is simple and easy to operate and simulate, but it is often compromised by light saturation. In order to overcome the shortcomings of those two methods, researchers began to invert LAI by using the red-edge band to construct a univariate red edge spectral vegetation index, which is between the red band and the near-infrared band [6]. This band is a good indicator of green plant growth and has a good correlation with LAI. Hansen et al. [7] demonstrated that the red-edge spectral vegetation index of 680–750 nm had a good correlation with wheat LAI. Anne et al. [8] explained that MTVI2 was more strongly correlated with NDVI and maize LAI. Darvishzah et al. [9] proved that REIP could estimate the LAI of maize effectively. Herrmann et al. [10] proved that REIP was more relevant than NDVI and maize LAI based on the Sentinel-2 image. Nahuel et al. [11] proved NDVIre than NDVI and MTVI2 were highly correlated with maize biomass based on the Rapid Eye image. Domestically, Cao et al. [12] thought that NDSI (940,730) inverts wheat LAI, which not only improves the inversion accuracy but also delays the saturation trend of the index. Su et al. [13] used NDSI (783, 705) univariate model. In this model, the performance of maize LAI proved that it could be used as the basis for judging the growth of maize. Xie et al. [14] used the TCARI/OSAVI double red-edge spectral vegetation index model to invert the winter wheat LAI, which proved that the model had a certain potential for inverting winter wheat LAI. Zheng et al. [15] used the CIre red-edge chlorophyll index to invert the winter wheat LAI, which proved that the model and biomass were highly accurate. Gao [16] proved that the hyperspectral red-edge parameter model can accurately predict cotton LAI. The above methods have delayed the phenomenon of spectral saturation compared with the traditional vegetation index NDVI, EVI, etc. However, the effectiveness of the delay is not enough to meet the requirement of inversion.

At present, because most satellite sensors do not include the red edge band, the research on whether the red edge spectral vegetation index can accurately reflect crop growth has certain limitations. Therefore, based on the red-edge bands data of Sentinel-2, the univariate red-edge spectral vegetation index can only provide partial information about crop growth conditions which are susceptible to the crop canopy structure, density and field climate. Therefore, a multi-variable red-edge spectral vegetation index is proposed to invert the winter wheat LAI in the booting stage based on the Sentinel-2 satellite MSI data and the univariate red-edge spectral vegetation index. This method can not only effectively delay the phenomenon of spectral saturation but also can provide the growth of winter wheat correctly at the booting stage and an accurate basis for monitoring crop growth as well.

## 2. Materials and Methods

### 2.1. Materials

#### 2.1.1. Study Area

The Sanmenxia City and Xinxiang City, Henan province (Figure 1) are selected as the study area to investigate the performance of LAI inversion algorithm for MSI data. The Sanmenxia study area is located in the Western Henan Plain. The annual average temperature is 12.2 °C, the annual precipitation is 528 mm, and the annual average frost-free period is 210 d. The Xinxiang County study area is located in the Eastern Henan Plain. The annual average temperature is 14 °C, and the annual average rainfall is 573.4 mm. The annual average frost-free period is 205 days. Both study areas are prevailing in northwest winds all year round. They are temperate continental monsoon climates with sufficient light and heat, and the rain and heat are in the same season, which is conducive to the growth of winter wheat.

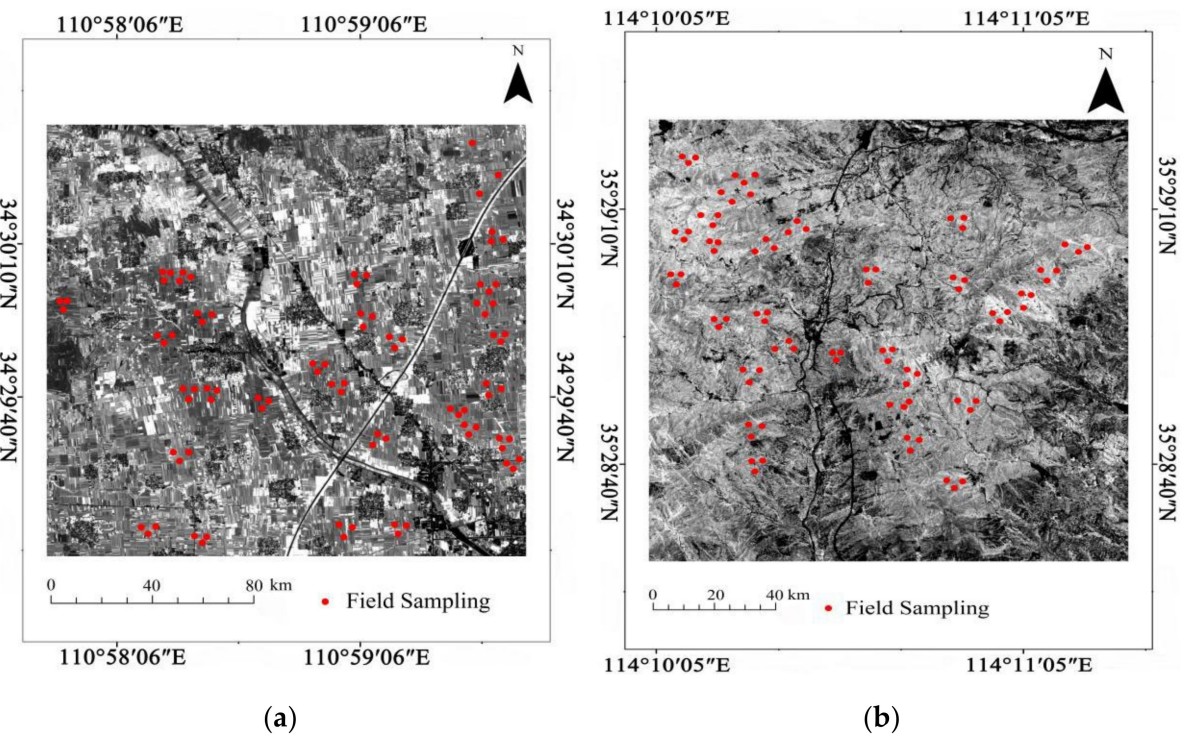

**(a)**　　　　　　　　　　　　　　　　　**(b)**

**Figure 1.** The left figure shows the geographical location of the research area in Henan province, (**a**) Sanmenxia study area, and (**b**) Xinxiang study area, the red points are LAI field survey locations.

### 2.1.2. Field LAI Measurements

LAI was measured by LAI-2000 plant canopy analyzer, and the collection date was 25 August 2021, which was almost synchronized with the satellite transit time. The daily collection time is 07:30 to 09:30 a.m. Fifty winter wheat quadrates with the size of $16 \times 16$ m were selected, and the average values of three sample points were taken as the LAI measurement results of the quadrates. Meanwhile, Huadai LT400 handheld GPS was used for field positioning of each sample point, and the positioning accuracy error was about $\pm 3$ m, indicating a high positioning accuracy. LAI-2000 plant canopy analyzer should avoid measurement under direct light as far as possible. During measurement, ABBBB was adopted. Firstly, A value above the canopy was measured against the sunlight, and then B value below the canopy was measured by the instrument close to the root of corn for four times.

### 2.1.3. Sentinel-2 MSI Data and Pre-Process

In order to improve the application ability of satellites, there are satellites carrying the red edge band, such as Rapid Eye satellite, Worldview-2 and Sentinel-2 satellite [17]. The Sentinel-2 satellite provides multi-spectral data with spatial resolutions of 10 m, 20 m, and 60 m, and contains three red-edge bands, which are very effective for inverting LAI and monitoring of crop growth. In this experiment, two sets of Sentinel-2 MSI data of Sanmenxia County and Xinxiang County were selected respectively, and the acquisition time of the images was 25 August 2021, which is basically the same as the LAI time of the real winter wheat. Image data come from the ESA data sharing website (http://scihub.copernicus.eu/dhus/#/home) accessed on 20 October 2021, which contains 13 bands [18], spatial resolution and central wavelength. Because the L1C image is orthocorrected, and the spatial resolution of each band is different, only atmospheric correction and resampling are needed for the image. Firstly, the image is atmospherically corrected by SNAP software, and then the image after atmospheric correction is resampled by the adjacent interpolation method so that the resolution of the resampled image band is 10 m.

## 2.2. Methods

### 2.2.1. Selection of Red-Edge Spectral Vegetation Index

Based on Sentinel-2 MSI data, the common univariate red-edge spectral vegetation index in LAI inversion of winter wheat was selected (Table 1), including The Red-edge Normalized Difference Spectral Indices 1 (NDSI1) and The Red-edge Normalized Difference Spectral Indices 2 (NDSI2), The Red-edge Normalized Difference Spectral Indices 3 (NDSI3), Modified Chlorophyll Absorption Ratio Index (MCARI), MERIS Terrestrial Chlorophyll Index (MTCI), Transformed Chlorophyll Absorption in Reflectance Index (TCARI), and Transformed Chlorophyll Absorption in Reflectance Index/the optimized soil adjusted vegetation index (TCARI/OSAVI). Based on the principle that band or band fusion contains a large amount of crop information, a multi-variable red-edge spectral vegetation index is constructed according to the above seven single-variable red-edge spectral vegetation indices (Table 1), which are NDSI1 + NDSI2, NDSI2 + NDSI3, NDSI1 + NDSI2 + NDSI3, NDSI1 + NDSI1 + NDSI2 + NDSI3,NDSI1 + NDSI2 + NDSI2 + NDSI2 + TCARI/OSAVI, NDSI1 + NDSI3 + TCARI/OSAVI,NDSI2 + NDSI3 + NDSI3 + TCARI/OSAVI, NDSI1 + NDSI2 + NDSI2 + NDSI3 + NDSI2 + NDSI3 + TCARI/OSAVI and NDSI1 + NDSI2 + NDSI3 + TCARI/OSAVI + MCARI + MTCI + TCARI.

**Table 1.** Red-Edge Spectral Vegetation Index.

| Variable Model | Red-Edge Spectral Vegetation Index | Formula | References |
|---|---|---|---|
| Univariate model | NDSI1 | $(R_{783} - R_{740})/(R_{783} + R_{740})$ | [19,20] |
| | NDSI2 | $(R_{740} - R_{705})/(R_{740} + R_{705})$ | [21] |
| | NDSI3 | $(R_{783} - R_{705})/(R_{783} - R_{705})$ | [22] |
| | MCARI | $[(R_{705} - R_{665}) - 0.2(R_{705} - R_{560})] \times (R_{705}/R_{665})$ | [23,24] |
| | MTCI | $(R_{665} - R_{705})/(R_{783} + R_{665})$ | [25,26] |
| | TCARI | $3[(R_{783} - R_{665}) - 0.2(R_{705} - R_{560})(R_{705}/R_{665})]$ | [27–29] |
| | TCARI/OSAVI | $3[(R_{783} - R_{665}) - 0.2(R_{705} - R_{560})(R_{705}/R_{665})]/(1 + 0.16)(R_{842} - R_{665})(R_{842} + R_{665} + 0.16)$ | [30,31] |
| Multivariate model | NDSI1 + NDSI2 | y = −3.138 + 5.4047NDSI1 + 5.5653NDSI2 | |
| | NDSI2 + NDSI3 | y = −1.333 + 5.500NDSI2 + 10.7176DNSI3 | |
| | NDSI1 + NDSI3 | y = −2.2846 + 5.7565NDSI1 + 11.1507DNSI3 | |
| | NDSI1 + NDSI2 + NDSI3 | y = −2.906 + 4.2064DNSI1 + 3.8653DNSI2 + 8.4752DNSI3 | |
| | NDSI1 + NDSI2 + TCARI/OSAVI | y = 1.974 + 3.5074NDSI1 + 3.518NDSI2 + 0.3093TCARI/OSAVI | |
| | NDSI1 + NDSI3 + TCARI/OSAVI | y = −1.561 + 3.923NDSI1 + 7.349DNSI3 + 0.2945TCARI/OSAVI | |
| | NDSI2 + NDSI3 + TCARI/OSAVI | y = −0.911 + 3.71DNSI2 + 6.9515DNSI3 + 0.3010TCARI/OSAVI | |
| | NDSI1 + NDSI2 + NDSI3 + TCARI/OSAVI | y = −2.0267 + 3.0019NDSI1 + 2.68NDSI2 + 6.0964NDSI3 + 0.2512TCARI/OSAVI | |
| | NDSI1 + NDSI2 + NDSI3 + TCARI/OSAVI + MCARI + MTCI + TCARI | y = 0.3475 + 1.0404NDSI1 + 0.6044NDSI2 − 0.4747NDSI3 − 0.0868TCARI + 2.1645MCARI + 1.8548MTCI + 0.2096TCARI/OSAVI | |

**Note:** R is spectral reflectance, with wavelengths of 783, 740, 705, 665, 560, and 842.

### 2.2.2. LAI Inversion Accuracy Assessment

Through precision evaluation, the fitting status of single and multi-variable red-edge spectral vegetation index with measured LAI can be effectively evaluated, and the optimal inversion model can be obtained. This paper chooses three methods to evaluate, namely, the determination coefficient ($R^2$), the root mean square error (RMSE) and the noise equivalent value (NE), as shown in Formulas (1)–(3).

$$R^2 = 1 - \frac{\sum_{i=1}^{n}(y - y_i)}{\sum_{c=1}^{n}(y - y_c)}. \tag{1}$$

$$RMSE = \sqrt{\frac{\sum_{i=1}^{n}(y_e - y_d)}{n}}. \tag{2}$$

$$NE = \frac{RMSE\{VI\ VS\ LAI\}}{d(VI)/d(LAI)} \tag{3}$$

In Formula (1), $y$ denotes the measured ecological parameters, $y_i$ denotes the estimated ecological parameters, and $y_c$ denotes the average value of the measured ecological parameters. In Formula (2), $y_i$ is the predicted value, $y_d$ is the measured value, and $n$ is the number of samples. In Formula (3), NE is the noise equivalent value of LAI, VI is the vegetation index, LAI is the measured leaf area index, RMSE {VIVSLAI} is the root mean square error in the fitting relationship between VI and LAI, and d(VI)/d(LAI) is the partial derivative of VI to LAI.

## 3. Results

### 3.1. Correlation Analysis

Before the regression fitting analysis, the correlation between the univariate and multivariate red-edge spectral vegetation index and the measured LAI of the winter wheat at the booting stage was firstly analyzed (Table 2). It can be seen from Table 2 that the red edge spectral vegetation index is highly correlated with the measured LAI of winter wheat at the booting stage, and the correlation coefficient is higher than 0.5994. Among them, the correlation coefficient between the multivariate red edge spectral vegetation index and the measured booting stage winter wheat LAI is significantly higher than the univariate red edge spectral vegetation index and the measured booting stage winter wheat LAI correlation coefficient and the correlation coefficient between NDSI1 + NDSI2 + NDSI3 + TCARI/OSAVI + MCARI + MTCI + TCARI and the measured LAI fitting of winter wheat at booting stage is optimal.

**Table 2.** Correlation analysis between red edge spectral vegetation index and measured LAI in winter wheat at the booting stage.

| Red-Edge Spectral Vegetation Index | Correlation Coefficient | |
| --- | --- | --- |
| | Sanmenxia County | Xinxiang County |
| NDSI1 | 0.7567 | 0.7660 |
| NDSI2 | 0.7585 | 0.7571 |
| NDSI3 | 0.7689 | 0.7536 |
| MCARI | 0.6007 | 0.5994 |
| MTCI | 0.7672 | 0.7478 |
| TCARI | 0.7487 | 0.7504 |
| TCARI/OSAVI | 0.6697 | 0.7091 |
| NDSI1 + NDSI2 | 0.8475 | 0.8507 |
| NDSI1 + NDSI3 | 0.8650 | 0.8692 |
| NDSI2 + NDSI3 | 0.8543 | 0.8774 |
| NDSI1 + NDSI2 + NDSI3 | 0.8974 | 0.8955 |
| NDSI1 + NDSI2 + TCARI/OSAVI | 0.7701 | 0.7694 |
| NDSI1 + NDSI3 + TCARI/OSAVI | 0.8859 | 0.8860 |
| NDSI2 + NDSI3 + TCARI/OSAVI | 0.8817 | 0.8758 |
| NDSI1 + NDSI2 + NDSI3 + TCARI/OSAVI | 0.9134 | 0.9120 |
| NDSI1 + NDSI2 + NDSI3 + TCARI/OSAVI + MCARI + MTCI + TCARI | 0.9149 | 0.9390 |

### 3.2. Red-Edge Spectral Vegetation Index and Measured LAI Fitting Analysis

We obtained the curves and linear regression models of the winter wheat LAI and 16 red-edge spectral vegetation indices in the experimental area of Sanmenxia and Xinxiang County, respectively, and adopted $R^2$ and RMSE. Accuracy analysis of the regression model and the results are shown in Figures 2 and 3.

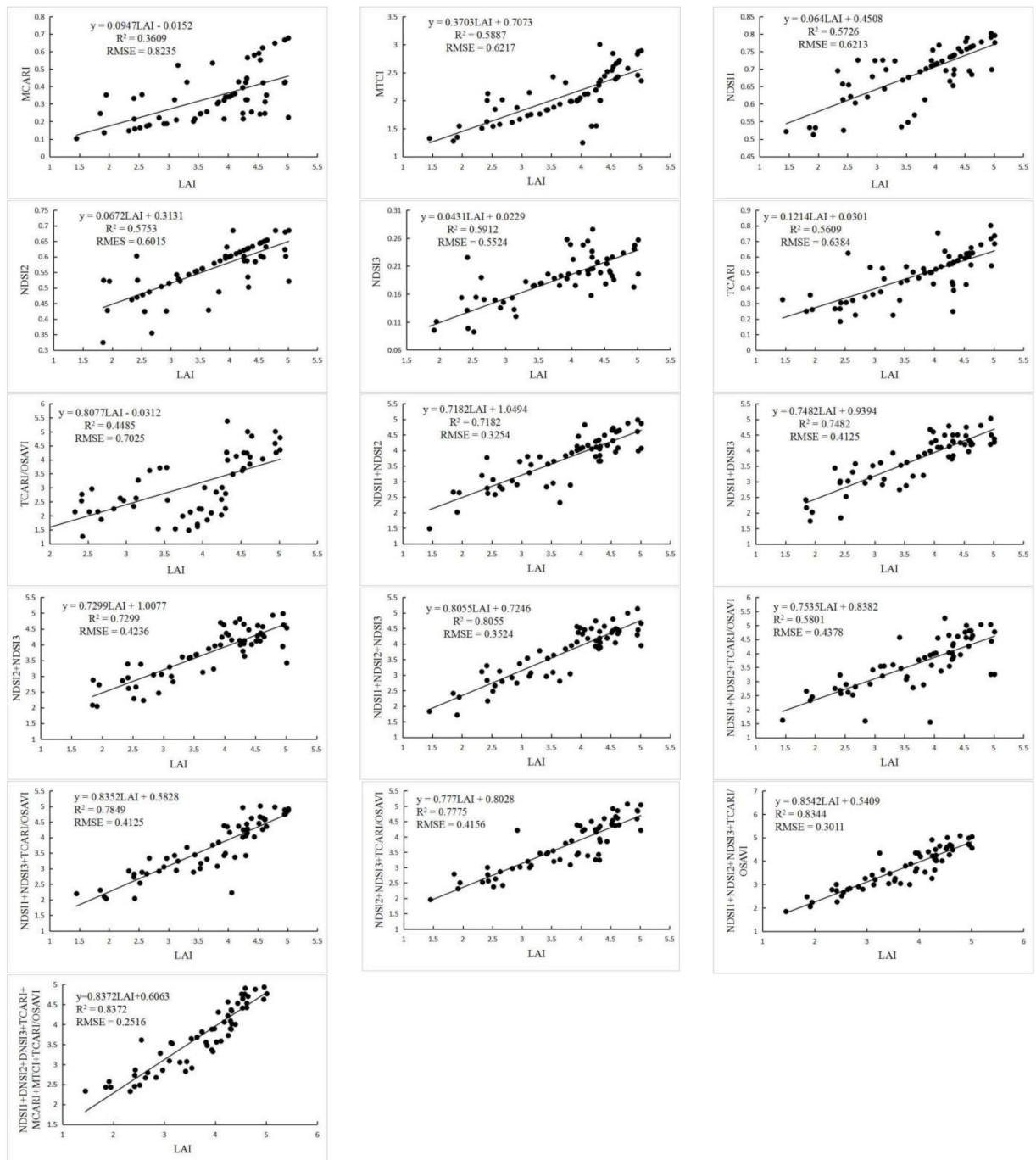

**Figure 2.** Fitting map of red-edge spectral vegetation index with Sanmenxia as an experimental area.

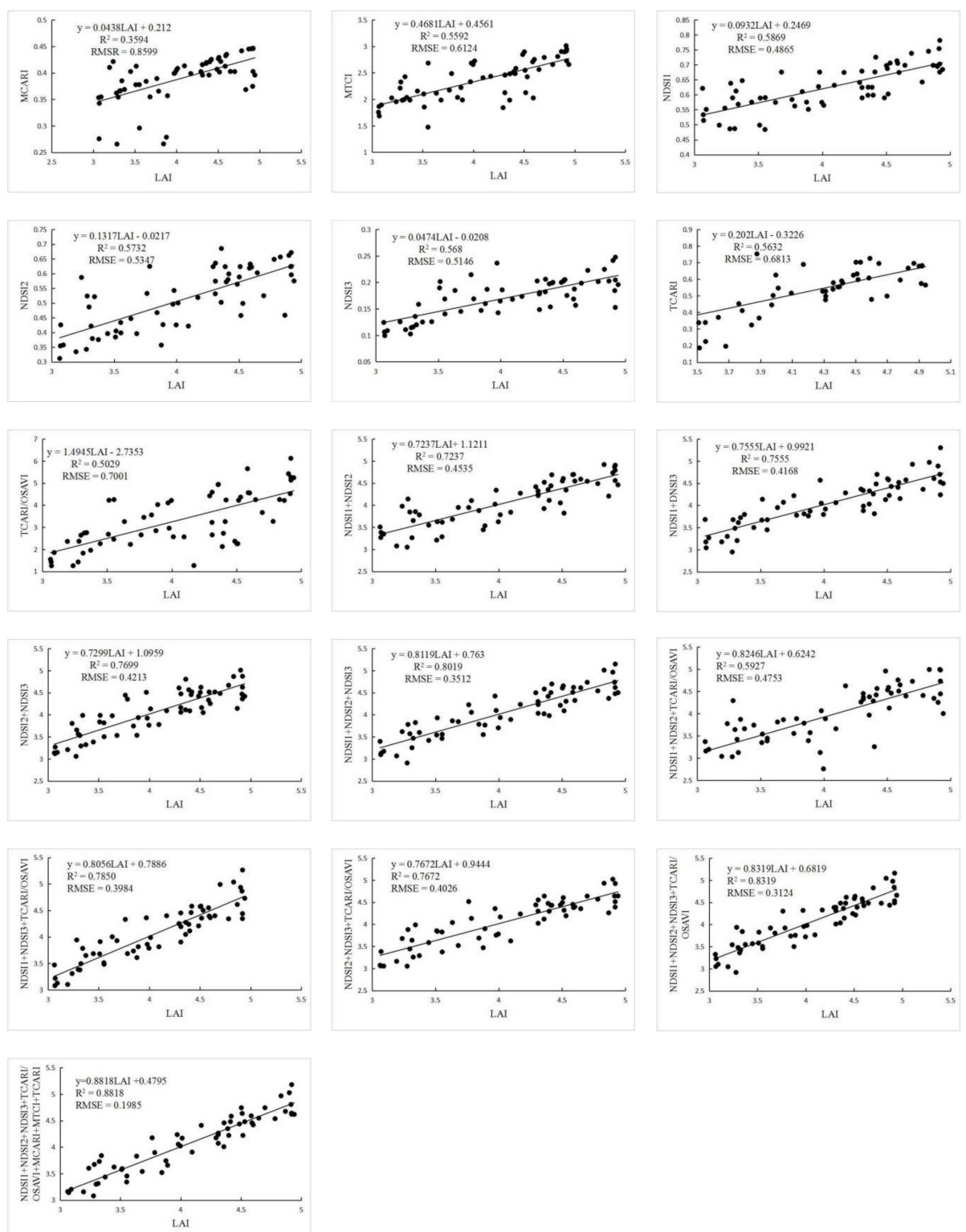

**Figure 3.** Fitting map of red-edge spectral vegetation index in Xinxiang County.

The test area of the univariate red edge spectral vegetation index, the model $R^2$ is above 0.3, and the RMSE is below 0.83 (Figure 2). Among them, NDSI3 has the optimal value, $R^2$ of 0.5912. The RMSE of 0.5524. In the multivariate red edge spectral vegetation index regression model, the model determination coefficient $R^2$ is above 0.7, and the RMSE is below 0.45. Among them, NDSI1 + NDSI2 + NDSI3 + TCARI/OSAVI + MCARI + MTCI

red edge spectral vegetation index and measured booting stage of winter wheat LAI have the best fitting accuracy, $R^2$ of 0.8372 and RMSE of 0.2516. It can be seen from Figure 3 that in Xinxiang County, the model $R^2$ is above 0.3 and the RMSE is below 0.86 in the univariate red-edge spectral vegetation index regression model. Among them, NDSI1 has the optimal value and $R^2$ of 0.5869, RMSE of 0.4865. In the multivariate red-edge spectral vegetation index regression model, the model $R^2$ is above 0.7 and the RMSE is below 0.45. Among them, NDSI1 + NDSI2 + NDSI3 + TCARI/OSAVI + MCARI + MTCI + TCARI multivariate red edge spectral vegetation index and measured the LAI fitting accuracy of winter wheat was the best at the booting stage, $R^2$ was 0.8818 and RMSE was 0.1985. According to the comprehensive analysis, the multivariate red-edge spectral vegetation index has a stronger ability to interpret the winter wheat LAI than the univariate red-edge spectral vegetation index.

### 3.3. Spectral Saturation Sensitivity Analysis

For the nonlinear fitting, only $R^2$ and RMSE are used to evaluate the correlation of the model, which will be affected by spectral saturation and produce some deviation. In order to further evaluate the performance of univariate and multivariate red-edge spectral vegetation index in the winter wheat LAI estimation, the noise equivalent value (NE) method was used to analyze the univariate and multivariate 16 red-edge spectral vegetation indices. The results are shown in Figure 4.

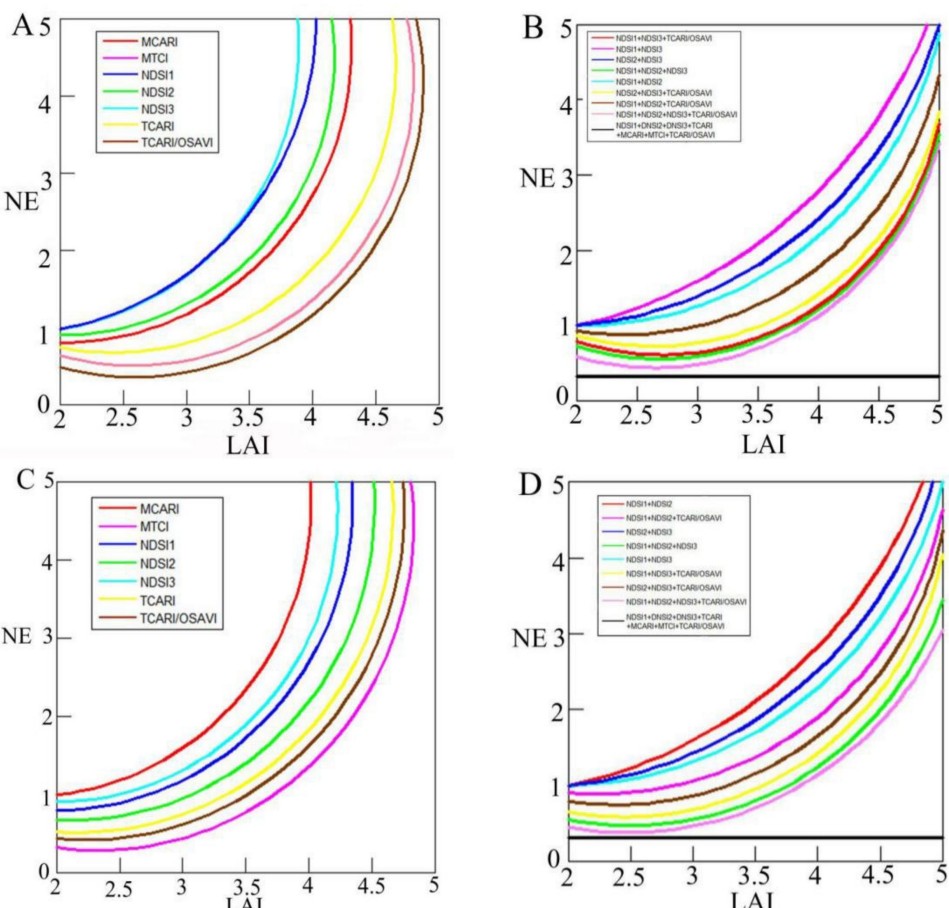

**Figure 4.** (**A**,**B**) are the sensitivity analysis of the red-edge spectral vegetation index in the experimental area of Sanmenxia County. (**C**,**D**) Sensitivity analysis of the red-edge spectral vegetation index in Xinxiang County.

As shown in Figure 4A, in the univariate red-edge spectral vegetation index regression model in Sanmenxia County as the experimental area, the sensitivity increases of MCARI, NDSI1, NDSI2, and NDSI3 are faster when the LAI is greater than 3.5; MTCI, TCARI, and TCARI/OSAVI have a faster rate of increase insensitivity when the LAI is greater than 4. It can be seen from Figure 4B that the NDSI1 + NDSI2 + NDSI3 + TCARI/OSAVI + MCARI + MTCI + TCARI red-edge spectral vegetation index is affected by the increase in the LAI value in the multivariate red edge spectral vegetation index regression model in Sanmenxia County as the experimental area. Other than the smaller, other red-edge spectral vegetation indexes, the sensitivity gradually begins to rise when the LAI is greater than 4.5. Figure 4C indicates that, in the univariate red-edge spectral vegetation index regression model with Xinxiang County as the experimental area, the sensitivity increases of MCARI, NDSI1 and NDSI3 are faster when the LAI is greater than 4; The sensitivity of MTCI, TCARI and TCARI/OSAVI increased rapidly when the LAI is greater than 4.5, NDSI2. It can be seen from Figure 4D that in the multivariate red edge spectral vegetation index regression model with Xinxiang County as the experimental area, except for NDSI1 + NDSI2 + NDSI3 + TCARI/OSAVI + MCARI + MTCI + TCARI multivariate red edge spectral vegetation index by LAI value In addition to the smaller influence, other multivariate red edge spectral vegetation indices gradually increase in sensitivity when the LAI is greater than 4.5. In general, the above 16 red edge spectral vegetation indices increased and increased with LAI, and the multivariate red edge spectral vegetation index delayed spectral saturation higher than the univariate red edge spectral vegetation index, of which NDSI1 + NDSI2 + NDSI3 + TCARI/OSAVI + MCARI + MTCI + TCARI spectral vegetation index it is the strongest to delay spectral saturation.

### 3.4. Model Verification

In order to test the reliability of the NDSI1 + NDSI2 + NDSI3 + TCAR/OSAVI + MCARI + MTCI + TCARI multivariate red edge spectral vegetation index to invert the winter wheat LAI in the booting stage, the inverted value of the winter wheat LAI in the booting stage was studied. Twenty-seven measured winter wheat LAI other than the sample were constructed for regression analysis and their accuracy was evaluated. Figure 5 is a fitting diagram of NDSI1 + NDSI2 + NDSI3 + TCARI/OSAVI + MCARI + MTCI + TCARI multivariate red edge spectral vegetation index prediction LAI and measured winter wheat LAI construction.

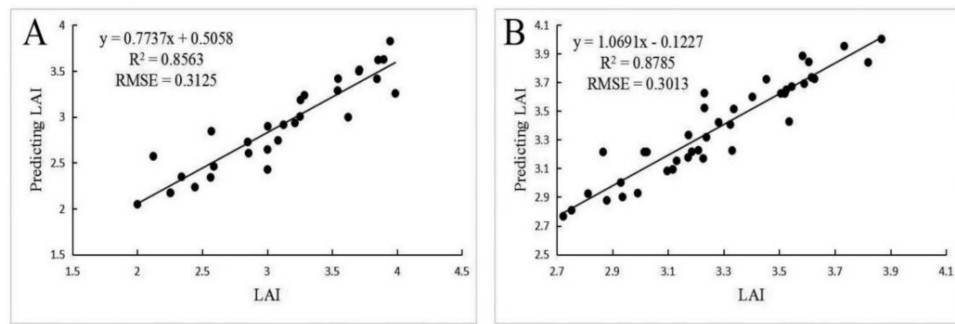

**Figure 5.** (**A**,**B**) verified fitting diagram of NDSI1 + NDSI2 + NDSI3 + TCARI/OSAVI + MCARI + MTCI + TCARI multivariate spectral vegetation index model by taking Sanmenxia and Xinxiang County as experimental areas, respectively.

As can be seen from Figure 5, in Sanmenxia County using NDSI1 + NDSI2 + NDSI3 + TCARI/OSAVI + MCARI + MTCI + TCARI multivariate red edge spectral vegetation index prediction LAI and measured LAI linear fitting, $R^2$ of 0.8563 and RMSE of 0.3125. In Xinxiang County as the experimental area and using NDSI1 + NDSI2 + NDSI3 + TCARI/OSAVI + MCARI + MTCI + TCARI multivariate red edge spectral vegetation index prediction, LAI and measured LAI linear fitting were $R^2$ of 0.8785 and RMSE of 0.3013. In summary, based on Sentinel-2 data, using NDSI1 + NDSI2 + NDSI3 + TCARI/OSAVI + MCARI + MTCI +

TCARI multivariate red edge spectral vegetation index to invert the winter wheat LAI in line with the growth of local winter wheat, and the inversion effect was better.

### 3.5. LAI Inversion Distribution Mapping

After verifying the accuracy of NDSI1 + NDSI2 + NDSI3 + TCARI/OSAVI + MCARI + MTCI + TCARI multivariate red-edge spectral vegetation index by the above methods, this paper adopted the model to invert and produce LAI spatial distribution maps of winter wheat in the incubation stage in Sanmenxia County and Xinxiang County, and the results are shown in Figure 6.

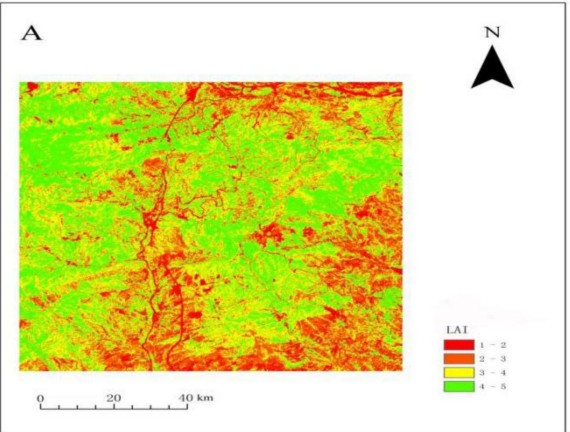
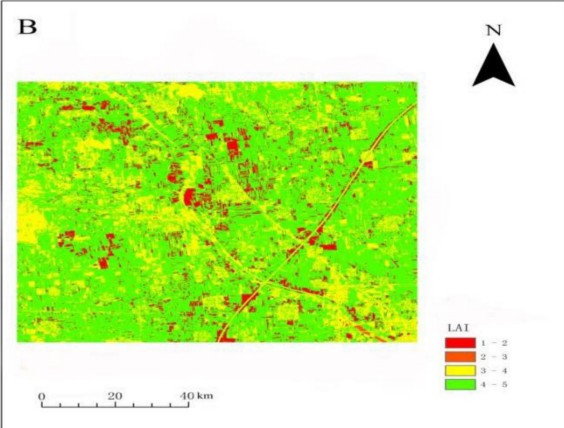

**Figure 6.** (**A**,**B**) are LAI distribution maps of winter wheat at the booting stage with Sanmenxia County and Xinxiang County as experimental areas, respectively.

As shown in Figure 6, LAI units are $m^2/m^2$. LAI values of winter wheat in the two experimental areas of Sanmenxia County and Xinxiang County were mainly between 4 and 5. Low LAI values of winter wheat are mainly concentrated in the vicinity of rivers, buildings, and roads, affecting the growth of winter wheat. High-value areas of winter wheat LAI are mainly concentrated in the local concentrated planting area of winter wheat.

### 4. Discussion

Affected by spectral saturation, the traditional vegetation index produces deviations in LAI prediction, which is mainly caused by the high reflection contrast between the red band and near-infrared band. The red-edge band is located at 680~760 nm and has an obvious reflectivity variation value from the lowest reflection point to the highest reflection point. The red-edge band is more sensitive to vegetation LAI than the visible band, and it is more effective in delaying spectral saturation. Therefore, a univariate and multivariate red-edge spectral vegetation index constructed by the red-edge band can effectively alleviate the problems discussed above. And the saturation delay of multivariate red-edge spectral vegetation index was higher than that of univariate red-edge spectral vegetation index, which was consistent with previous research results [14].

Inversion analysis of winter wheat LAI at the booting stage is carried out by 7 univariate red-edge spectral vegetation indexes and 9 multivariate red-edge spectral vegetation indexes constructed based on the red-edge band. Correlation and fitting analysis show that the LAI results of winter wheat inversion by univariate red-edge spectral vegetation index are generally low, and there are singular points. The inversion results of multivariate red-edge spectral vegetation index are uniformly distributed on both sides of the regression line, show red spectral vegetation index of correlation and multivariate fitting accuracy is higher than single variable refers to the red-edge spectral vegetation, and the inversion results are closer to the actual value, among them, the NDSI1 + NDSI2 + NDSI3 + TCARI MCARI/OSAVI + MCARI + MTCI + TCARI and multivariate red-edge spectral vegetation

index contains the highest correlation coefficient and fitting precision. which was consistent with previous research results [15].

At the same time, sensitivity analysis found that when winter wheat LAI was less than 3.5, the single variable red-edge spectral vegetation index was more sensitive to winter wheat LAI, but when winter wheat LAI was more than 3.5, it was found that the index was prone to spectral saturation in winter wheat LAI inversion, which made the prediction value of winter wheat LAI lower. Therefore, a univariate red-edge spectral vegetation index is not suitable for LAI inversion of winter wheat at the booting stage. In the multivariate red-edge spectral vegetation index, except for NDSI1 + NDSI2 + NDSI3 + TCARI/OSAVI + MCARI + MTCI + TCARI, other multivariate red-edge spectral vegetation indices have higher sensitivity when LAI of winter wheat is less than 4.5. However, when LAI of winter wheat is greater than 4.5, the problem of spectral saturation also arises. Although the phenomenon of spectral saturation is delayed compared with the univariate red-edge spectral index, it is also not suitable to LAI inversion of winter wheat at the booting stage. Only NDSI1 + NDSI2 + NDSI3 + TCARI/OSAVI + MCARI + MTCI + TCARI maintained high sensitivity and predictability for winter wheat LAI at booting stage. The model can accurately invert winter wheat LAI at the booting stage, which was consistent with previous research results [15,16].

A comprehensive comparison of 16 red-edge spectral vegetation indices shows that NDSI1 + NDSI2 + NDSI3 + TCARI/OSAVI + MCARI + MTCI + TCARI multivariate red-edge spectral vegetation index inversion accuracy of winter wheat LAI at booting stage is the highest among 16, and completely coincides with the growth status of winter wheat at booting stage. It shows that this model can more accurately invert the growth trend of winter wheat at the booting stage, in light of the Sentinel-2 image.

## 5. Conclusions

The main purpose of this paper is to explore the performance of the multivariate red-edge spectral vegetation index proposed in the estimation of winter wheat growth at the booting stage. Based on this, the univariate and multivariate red-edge spectral vegetation indices of Sentinel-2 data extraction were used to establish regression equations with the measured LAI of winter wheat in the booting stage, and the univariate and multivariate red-edge spectral vegetation index and the measured winter wheat LAI were analyzed. The findings, therefore, conclude as follows:

(1)　The univariate red-edge spectral vegetation index has a certain correlation with the measured LAI of winter wheat at the booting stage, but weaker than the multivariate red-edge spectral vegetation index.

(2)　The fitting accuracy of the multivariate red-edge spectral vegetation index was higher than that of the univariate red-edge spectral vegetation index, and the more fitting variables, the better the correlation, the higher the inversion precision, the better the inversion effect. Among them, NDSI1 + NDSI2 + NDSI3 + TCARI/OSAVI + MCARI + MTCI + TCARI multivariate red-edge spectral vegetation index and the measured LAI fitting accuracy of the winter wheat were the highest, which proved that the model could meet the requirements of LAI inversion accuracy of winter wheat.

(3)　With the increase in LAI, the spectral saturation increase rate of multivariate red-edge spectral vegetation index was slower than that of univariate red-edge spectral vegetation index, because the combination of red-edge band delayed the spectral saturation effect to some extent.

(4)　The red-edge spectral vegetation index constructed by the red-edge band in the Sentinel-2 remote sensing image had a strong inversion ability for winter wheat LAI at the booting stage.

This study explored the advantages of the Sentinel-2 data, including the red-edge band inversion of winter wheat LAI, but the two models were based on limited maize field measurements in a single test area, which made the model somewhat limited. Future

research will inevitably apply the model to multiple test areas and multiple species in order to enhance the universality of the model.

**Author Contributions:** Acquisition of the financial support for the project leading to this publication, X.W., G.C. and Q.Z. Application of statistical, mathematical, computational, or other formal techniques to analyze or synthesize study data, X.Z., Z.Y. and X.L. Preparation, creation, and/or presentation of the published work by those from the original research group, specifically critical review, commentary, or revision, including pre- or post-publication stages, G.C. and X.W. All authors have read and agreed to the published version of the manuscript.

**Funding:** This research was funded by 2016 National Key Research and Development Plan, grant number 2016YFC0803103, research on key technology of agricultural remote sensing monitoring, and Henan Provincial University Innovation Team Support Plan, grant number 14IRTSTHN026.

**Institutional Review Board Statement:** Not applicable.

**Informed Consent Statement:** Informed consent was obtained from all subjects involved in the study.

**Data Availability Statement:** The code used in this study is available by contacting the corresponding author.

**Acknowledgments:** The authors thank by Jinrui Fan, and Mengen Wang for the assistance with flight tests, and with Zhengfang Lou, and Lu Wang for assistance LAI measurements.

**Conflicts of Interest:** The authors declare no conflict of interest.

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
