# Peer review of "Inversion of Wheat Leaf Area Index by Multivariate Red-Edge Spectral Vegetation Index"

_sustainability, doi:10.3390/su142315875_

Round 1

Reviewer 1 Report

1. Introduction need to be more improved. First paragraph should be about problem statement then focus on the study area. Why this study is required. How the previous studies tried to handle this issue?. At the end of the Introduction section, authors should give some important points: what were the problems with the previous studies? What is the research gap? How did you try to fill these gaps? What was your contribution?

3)  2. 2. There are some grammatical errors, English language need to be improved. Please check the manuscript carefully

3. what is the benefit from applying the current study and show the differences with papers mentioned above?. Also try to include them in this section to enhance it.

4)   Please improve the quality of all figures in the manuscript

5) There is no discussion with previous works to appear and compare between results obtained and previous findings of researchers. Please include that in paper to promote your results

6)     Please the conclusion section should be shortened. It is very long. Also add simple sentences about the outlook. How to improve this study (future work).

3)   

Author Response

  1. Introduction need to be more improved. First paragraph should be about problem statement then focus on the study area. Why this study is required. How the previous studies tried to handle this issue?. At the end of the Introduction section, authors should give some important points: what were the problems with the previous studies? What is the research gap? How did you try to fill these gaps? What was your contribution?

Answer: At present, because most satellite sensors do not include the red edge band, the research on whether the red edge spectral vegetation index can accurately reflect crop growth has certain limitations. Therefore, based on the red-edge bands data of Sentinel-2, the univariate red-edge spectral vegetation index can only provide partial information about crop growth conditions which are susceptible to the crop canopy structure, density and field climate. Therefore, a multi-variable red-edge spectral vegetation index is proposed to invert the winter wheat LAI in the booting stage based on the Sentinel-2 satellite MSI data and the univariate red-edge spectral vegetation index. This method can not only effectively delay the phenomenon of spectral saturation, but also can provide the growth of winter wheat correctly at the booting stage and an accurate basis for monitoring crop growth as well.

  1. There are some grammatical errors, English language need to be improved. Please check the manuscript carefully

Answer: The author has modified it as required.

  1. what is the benefit from applying the current study and show the differences with papers ment

Answer: This method can not only effectively delay the phenomenon of spectral saturation, but also can provide the growth of winter wheat correctly at the booting stage and an accurate basis for monitoring crop growth as well.ioned above?. Also try to include them in this section to enhance it.

  1. Please improve the quality of all figures in the manuscript

Answer: The authors have improved the quality of the legends in Figures 2, 3, 4, and 5.

  1. There is no discussion with previous works to appear and compare between results obtained and previous findings of researchers. Please include that in paper to promote your results

Answer: The author has modified it as required.

  1. Please the conclusion section should be shortened. It is very long. Also add simple sentences about the outlook. How to improve this study (future work).

Answer: The article has separated the discussion from the conclusion. The conclusion has been modified as required. This study explored the advantages of the Sentinel-2 data including the red-edge band inversion of winter wheat LAI, but the two models were based on limited maize field measurements in a single test area, which made the model somewhat limited. Future research will inevitably apply the model to multiple test areas and multiple species in order to enhance the universality of the model.

Reviewer 2 Report

The authors have constructed both univariate and multivariate red-edge spectral vegetation index regression models using Sentinel-2 satellite data. The concept is well demonstrated using the measured LAI over summer maize crop and further establishing relationships among satellite and measured data. I find there are some issues that need to address. The specific comments are given below. Accordingly, a major revision of the manuscript has been recommended.

Major Comments:

1)      Abstract: The methodology part was dominated rather than key findings. Revise accordingly and also mention the best R2 and RMSE.

2)      Introduction: it was well written.  But some places need rewriting like: L83-L85 (move this part to the data description).

3)      Section 2.1.1.: mention summer maize growing months?

4)      Section 2.1.1.: mention the number of LAI measurements?

5)      Table 1: Delete this. Write one sentence about the resolution inside the text. Only mention about bands that were used in this study.

6)      Fig 2 & 3: Improve Resolution. In the caption write what you mean by dash lines. Is the regression model are random selection like linear, exponential, log so on ….Justify the selection

7)      Fig 6: write the unit of LAI and also inside the text

8)      Discussion: It is a kind of summary of results but I recommend rewriting this section to justify how these results are in harmony with the literature

9)       Address the limitation of the study under discussion.

10)  Check English through as I find a few incomplete sentences

Minor Comments:

1.      L17: Rewrite & Check the spell of "thar"

2.      L21: Check the incomplete sentence "summer maize LAI in the"

3.      L63: Check NDVIre

4.      L82: remove the repeated term satellite

5.      L191: simply write R2 without full abbreviation here

6.      Many abbreviations are not given in their first-time use (like MSI & so on  ….)

Author Response

The authors have constructed both univariate and multivariate red-edge spectral vegetation index regression models using Sentinel-2 satellite data. The concept is well demonstrated using the measured LAI over summer maize crop and further establishing relationships among satellite and measured data. I find there are some issues that need to address. The specific comments are given below. Accordingly, a major revision of the manuscript has been recommended.

Major Comments:

  • Abstract: The methodology part was dominated rather than key findings. Revise accordingly and also mention the best R2 and RMSE.

Answer: The results showed that the multivariable red-edge spectral vegetation index of NDSI1+NDSI2+NDSI3+TCARI/OSAVI+MCARI+MTCI+TCARI was the best model for inverting the winter wheat LAI. The R2, the RMSE and the NE value were all satisfied the requirements of the inversion precision (R2 = 0.8372/0.8818, RMSE= 0.2518/0.1985, NE=5/5).

  • Introduction: it was well written.  But some places need rewriting like: L83-L85 (move this part to the data description).

Answer: The author has modified it as required.

  • Section 2.1.1.: mention summer maize growing months?

Answer:Section 2.1.2 mention LAI measurement and measurement time.

  • Section 2.1.1.: mention the number of LAI measurements?

Answer:Section 2.1.2 mention LAI measurement and measurement time.

  • Table 1: Delete this. Write one sentence about the resolution inside the text. Only mention about bands that were used in this study.

Answer: The author has modified it as required.

  • Fig 2 & 3: Improve Resolution. In the caption write what you mean by dash lines. Is the regression model are random selection like linear, exponential, log so on ….Justify the selection

Answer:The authors have improved the quality of the legends in Figures 2, 3, 4, and 5.

  • Fig 6: write the unit of LAI and also inside the text

Answer:The author has modified it as required. LAI units are m2/m2.

  • Discussion: It is a kind of summary of results but I recommend rewriting this section to justify how these results are in harmony with the literature.

Answer:The article has separated the discussion from the conclusion. The conclusion has been modified as required.

  • Address the limitation of the study under discussion.

Answer:The conclusion contains limitations. This study explored the advantages of the Sentinel-2 data including the red-edge band inversion of winter wheat LAI, but the two models were based on limited maize field measurements in a single test area, which made the model somewhat limited. Future research will inevitably apply the model to multiple test areas and multiple species in order to enhance the universality of the model.

  • Check English through as I find a few incomplete sentences

Answer: The author has modified it as required.

Minor Comments:

  1. L17: Rewrite & Check the spell of "thar"

Answer: Thar has been modified to that.

  1. L21: Check the incomplete sentence "summer maize LAI in the"

Answer: In this study, the Sentinel-2 data were used to invert the winter wheat LAI.

  1. L63: Check NDVIre

Answer: The author has modified it as required.

  1. L82: remove the repeated term satellite

Answer: The author has modified it as required.

  1. L191: simply write R2 without full abbreviation here

Answer: The author has modified it as required.

Reviewer 3 Report

The conclusion section is too long and therefore should be summarized with the important findings from the study and possible recommendations.

Improve the quality of Figure 2, 3, 4, and 5 legends are difficult to read in the current state

Author Response

1.The conclusion section is too long and therefore should be summarized with the important findings from the study and possible recommendations.

Answer:The article has separated the discussion from the conclusion. The conclusion has been modified as required.

2.Improve the quality of Figure 2, 3, 4, and 5 legends are difficult to read in the current state.

Answer:The authors have improved the quality of the legends in Figures 2, 3, 4, and 5.

Round 2

Reviewer 2 Report

The authors have improved the overall quality of the manuscript and as suggested they have answered every question. However, fix one issue before the acceptance. Tables orders are not correct. It started with Table 2. 

Author Response

The authors have improved the overall quality of the manuscript and as suggested they have answered every question. However, fix one issue before the acceptance. Tables orders are not correct. It started with Table 2.

Answer: The author has modified it as required.